

# Highly time-resolved chemical characteristics and aging process of submicron aerosols over the central Himalayas

Yishen Wang[1], Yanqing An[2], Yulong Tan[2], Kemei Li[2], Jianzhong Xu[1], Shugui Hou[1]

[1]School of Oceanography, Shanghai Jiao Tong University, Shanghai, 200030, China

[2]Northwest Institute of Eco-Environment and Resources, Chinese Academy of Sciences, Lanzhou 730000, China

*Correspondence to*: Jianzhong Xu (jzxu78@sjtu.edu.cn)

**Abstract.** Aerosol particles transported from South Asia, especially biomass burning (BB) emission related aerosols during pre-monsoon, have significant climate effect in the Himalayas. The details on complicated physicochemical properties and aging process of aerosols are important for understanding this climate effect. An Aerodyne high-resolution time-of-flight

aerosol mass spectrometer co-located with gas analyzers was deployed during 25 April 2022 to 25 May 2022 to study the highly time-resolved chemical characteristics and aging process of submicron aerosols ($PM_1$) on the northern slope of the Himalayas. The 10-min resolution mass concentration of $PM_1$ varied from 0.1 to 12.2 µg m$^{-3}$ during this study, with an average of 1.7 ± 1.6 µg m$^{-3}$. Organic aerosols (OA) showed a dominant contribution (46.2 %) to $PM_1$ following by sulfate (20.8 %), BC (19.4 %), ammonium (8.5 %), nitrate (4.8 %) and chloride (0.4 %). Evolution of bulk OA in the $f$44 vs. $f$60

space showed clear aging process from less aged BB plumes to highly oxidized state in polluted period. Positive matrix factorization (PMF) on the high-resolution organic mass spectra resolved two oxygenated OA (OOA) factors, i.e., a less-oxidized OOA influenced by biomass burning (OOA-BB) and a more-oxidized OOA (MO-OOA). We performed a case study to explore the OOA formation mechanism during long-range transport. The results indicated aqueous-phase process and photochemical reaction together elevated OOA concentrations and ageing processing, consistent with secondary

inorganic aerosol production. This study underscores the significant occurrence of BB aerosols in Himalayas and provides insights into the oxidative processing in this remote region.

## 1 Introduction

Aerosol particles, as an important component of the atmosphere, have large impacts on both climate and human health (Garrett and Zhao, 2006; Landrigan et al., 2018). The Himalayas were once considered as a natural barrier to isolate air

pollution from densely populated South Asia that made the Tibetan Plateau (TP) one of the most pristine regions. However, growing evidences including isotopic fingerprinting from different environmental media across the TP (Huang et al., 2023; Li et al., 2016; Wang et al., 2025; Zhang et al., 2024) and continuous atmospheric observations (Bonasoni et al., 2010), have demonstrated that polluted air masses from South Asia could be transported across the Himalayas. Ground-based and satellite observations, in combination with model simulations, suggested that during the pre-monsoon season, atmospheric



pollutants stacked up in the southern foothills of the Himalayas, then climbed over the Himalayas via valley wind circulation and topographic lifting, and eventually transported to the inland of the TP (Lüthi et al., 2015). Consequently, trans-Himalayan transport of South Asian emissions and their associated environmental impacts over the TP have drawn growing scientific attention (Zhao et al., 2019).

Knowledge of aerosol chemical characterization and sources is crucial for assessing their climate impacts. A number of field

measurements have been carried out to investigate aerosols in the Himalayas. Yu et al. (2024) found that organic matter contributed to ~70 % of the total aerosol mass and secondary organic matter was the major aerosol component at Yadong on the south slope of the Himalayas. Biomass burning (BB) has been considered as an important source of OA in the Himalayas, especially in the pre-monsoon period, as indicated by elemental and molecular tracers (Arun et al., 2021; Cong et al., 2015; Stone et al., 2010). The optical properties of brown carbon (BrC) over the Himalayas exhibited stronger light absorption

compared to the northeastern TP due to the significant contribution from BB emissions (Xu et al., 2020; Xu et al., 2022). An et al. (2019) reported high fraction of nitrogen-containing species in WSOC, and identified directly molecular evidences of biomass-burning emitted compounds using electrospray ionization Fourier transform-ion cyclotron resonance mass spectrometry (ESI-FTICR MS). These transported aerosols from South Asia, especially OA from BB emission, have been suggested to heat the upper troposphere over the TP, with implications for the South Asian monsoon system and contributing

to accelerated glacial melt in the Himalayas (Lau et al., 2006; Zhang et al., 2015b). Therefore, there is an urgent need to capture the detailed information of OA aging process and secondary formation in BB emission events, which requires highly time-resolved in-situ observation.

The Aerodyne aerosol mass spectrometers (AMS) have been widely used to characterize submicron aerosols (PM$_1$), with advantages in high time-resolution and determination of OA factors based on mass spectrum data (Canagaratna et al., 2004;

Sun et al., 2018b). Furthermore, synchronous real-time trace gas observation is helpful to understand phase partitioning and secondary OA (SOA) formation processes with AMS results (Duan et al., 2022; Hu et al., 2016; Rogers et al., 2025). Studies based on AMS measurements are relatively rare and new in the TP limited by harsh natural conditions, such as Nam Co in the central TP (Wang et al., 2017a; Xu et al., 2018), Mt. Yulong and Motuo in the southeastern TP (Xu et al., 2024; Zheng et al., 2017), Laohugou and Waliguan at the northeastern TP (Xu et al., 2024; Zhang et al., 2019). Zhang et al. (2018) deployed

a high-resolution time-of-flight AMS (HR-ToF-AMS) at the Qomolangma Station for Atmospheric and Environmental Observation and Research (QOMS) on the northern slope of the Himalayas, analyzing the chemical evolution of aerosol characteristics based on a typical BB plume. However, the only previous AMS study in the Himalayas lacks a better understanding of the OA evolution combining with gas observation and in-depth discussion on the formation mechanism of SOA.

In this study, we conducted intensive real-time field measurements of both gas and submicron aerosols at QOMS in the pre-monsoon period using HR-ToF-AMS and gas analyzers. We examined the chemical composition and size distributions of aerosols and identified transport pathways via trajectory model and concentration-weighted trajectory (CWT) method combining with satellite data. Meanwhile, OA factors were resolved using positive matrix factorization (PMF) analysis



based on high-resolution mass spectrum and then discussed the OA evolution. The main objectives of our study were to
describe OA oxidation pathways and SOA formation mechanism under the influence of biomass burning emission in the
Himalayas.

# 2 Methodology

## 2.1 Sampling site and measurements

Non-refractory submicron aerosols (NR-PM$_1$) were measured by a HR-ToF-AMS (Aerodyne Research Inc., Billerica, MA,
USA) at QOMS, which is located on the northern slope of Mt. Everest (28.36° N, 86.95° E; 4276 m a.s.l.; Fig. 1) from 25
April 2022 to 25 May 2022. The QOMS and its surroundings are a pristine region except for a small village that is about 10
km south of the station. The land surface is covered by sand and gravel with sparse vegetation. The regional atmospheric
circulation at the QOMS is controlled by the westerlies and South Asian summer monsoon. The mountain-valley circulation
dominates the diurnal variation of wind direction due to distinct thermal forcing from the southern mountains and glaciers.
Detailed descriptions of QOMS can be found in several previous studies (Ma et al., 2023).

The HR-ToF-AMS has been described elsewhere (DeCarlo et al., 2006). Briefly, particles are sampled into the instrument
through a critical orifice, accelerated into the sizing vacuum chamber, and directed onto a resistively heated surface (600 °C)
under a high vacuum and ionized by a 70 eV electron impact, before finally being detected by a high-resolution mass
spectrometer. A detailed summary is given in Text S1. The NR-PM$_1$ species (organics, sulfate, nitrate, ammonium, and
chloride) were measured with the V-mode, including the mass spectrum (MS) mode and the particle P-ToF mode. The time
resolution of AMS data is 10 min before 9 May and 5 min after that. Black carbon was measured by an Aethalometer (model
AE33, Magee Scientific Corp., Berkeley, CA, USA), and gaseous carbon oxide (CO), ozone (O$_3$), sulfur dioxide (SO$_2$),
ammonia (NH$_3$), as well as nitrogen oxides (NO$_x$) were measured by Thermo gas analyzers (Model 48i, 49i, 43i, 17i and 42i,
respectively, Thermo Fisher Scientific, Waltham, MA, USA). The meteorological parameters including solar radiation,
relative humidity (RH), temperature, wind speed, and wind direction were measured using a Vantage Pro2 weather station
(Davis Instruments Corp., Hayward, CA, USA).

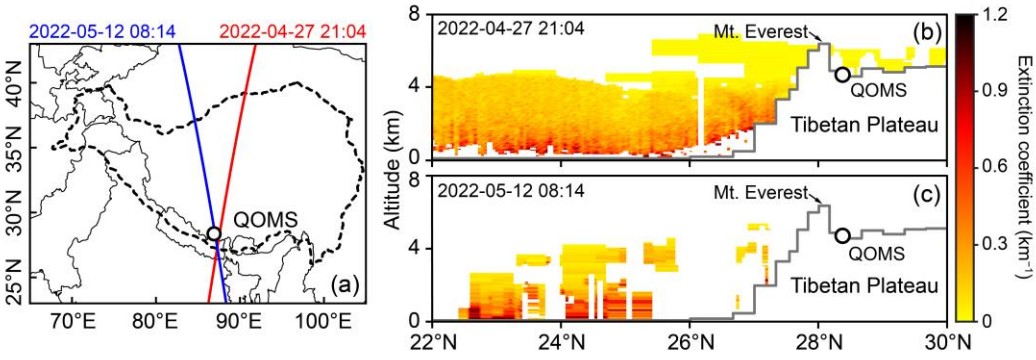



**Figure 1.** The location of the QOMS and aerosol vertical distribution over the southern TP and southern Asia in observation period. The orbit tracks of CALIPSO **(a)** and vertical distributions of the aerosol extinction coefficient at 532 nm along the CALIPSO tracks **(b, c)** on 27 April and 12 May 2022. Dashed black lines in (a) represent the Tibetan Plateau.

## 2.2 HR-ToF-AMS data analysis

The HR-ToF-AMS data were analyzed using the standard AMS analysis software SQUIRREL (v1.66) for unit resolution mass spectrum data and PIKA (v1.26) for high-resolution mass spectra data (HRMS) written in Igor Pro (WaveMetrics, Inc., Oregon USA). An empirical particle collection efficiency (CE) of 0.5 was used in this study based on the fact that a Nafion dryer was installed in front of the inlet to keep RH below 30 % and aerosols were approximately neutralized (see Sect. 3.1 and Fig. S2 for details). The relative ionization efficiency (RIE) values for organics, nitrate, sulfate, ammonium and chloride were set as the default values, i.e., 1.4, 1.1, 1.2, 4.0 and 1.3, respectively (Canagaratna et al., 2007). The elemental ratios of OA (O/C, H/C and N/C) were determined using the Improved-Ambient method (Canagaratna et al., 2015). PMF with the PMF2.exe algorithm (Paatero and Tapper, 1994) was applied on the HRMS to resolve potential OA components with different sources and processes. The PMF results were then evaluated using an Igor Pro-based PMF Evaluation Tool (PET, v3.08) (Ulbrich et al., 2009) following procedures in Zhang et al. (2011). After careful evaluation of the mass spectra and time series of OA factors, a 2-factor solution with the rotational parameter (fpeak) = 0 was chosen.

## 2.3 Air mass back trajectory analysis

The Hybrid Single-Particle Lagrangian Integrated Trajectory (HYSPLIT) model was used to calculate 3-day backward trajectories (Stein et al., 2015). The Global Data Assimilation System meteorological dataset (1° × 1°) was input to the model as the background meteorological data. The initial height was set at 500 m above the QOMS. The concentration-weighted trajectory (CWT) with the grid precision of 0.25° × 0.25° was then applied to identify potential source regions of $PM_1$ species (Hsu et al., 2003). The CWT is expressed as:

$$C_{ij} = \frac{\sum_{l=1}^{M} C_l \tau_{ijl}}{\sum_{l=1}^{M} \tau_{ijl}} \cdot W_{ij} \tag{1}$$

$$W_{ij} = \begin{cases} 1 & 3n_{ave} < n_{ij} \\ 0.7 & 1.5n_{ave} < n_{ij} \leq 3n_{ave} \\ 0.42 & n_{ave} < n_{ij} \leq 1.5n_{ave} \\ 0.05 & n_{ij} \leq n_{ave} \end{cases} \tag{2}$$

where $C_{ij}$ is the weighted average concentration in the grid cell (i, j), $M$ is the total number of trajectories, $C_l$ and $\tau_{ijl}$ are the NR-$PM_{2.5}$ concentration and the residence time of the corresponding trajectory (l) in the grid cell (i, j), $W_{ij}$ is the weight function, $n_{ave}$ is the average number of trajectories in all grid cells, $n_{ij}$ is the number of trajectories in the grid cell (i, j). Note



that the grid cells with no trajectories were excluded. The parameters for $W_{ij}$ refer to Dai et al. (2021) and Guirado et al. (2014).

## 2.4 Other relevant data

The Extended Aerosols Inorganics Model (E-AIM; http://www.aim.env.uea.ac.uk/aim/aim.php) was used to calculate aerosol liquid water content (ALWC). The input data included the concentrations of sulfate, nitrate, ammonium and chloride measured by the HR-ToF-AMS as well as the temperature and relative humidity of ambient air. Fire hotspots data over South Asia was obtained from the Fire Information for Resource Management System (FIRMS) provided by the MODIS satellite with 1-km resolution (https://firms.modaps.eosdis.nasa.gov). Aerosol optical depth (AOD) at 550 nm was sourced from the Modern-Era Retrospective analysis for Research and Applications version 2 (MERRA-2) with a resolution of $0.5° \times 0.625°$ (https://disc.gsfc.nasa.gov/datasets/M2T1NXAER_5.12.4/summary). The westerlies index, defined as the 500-hPa zonal wind at QOMS, and India-Burma Trough index, defined as the 700-hPa geopotential height over the region (17.5° N–27.5° N, 80° E–100° N) were calculated to illustrate the variation of atmospheric circulation at QOMS. The zonal wind and geopotential height data were extracted from the European Center for Medium-Range Weather Forecasts (ECMWF) ERA5 datasets in a grid of $0.25° \times 0.25°$ (https://cds.climate.copernicus.eu/datasets/reanalysis-era5-pressure-levels?tab=overview).

## 3 Results and discussion

### 3.1 Overview of the Campaign

#### 3.1.1 Mass concentration and chemical composition of $PM_1$

An overview of mass concentrations and compositions of $PM_1$ (NR-$PM_1$ + BC) as well as meteorological conditions is shown in Fig. 2. The air temperature ranged from −5.4 to 19.8 °C with an average of $7.1 \pm 4.9$ °C, and the RH ranged from 6.2 to 92.0 % with an average of $55.8 \pm 16.9$ %. The air mass at QOMS was predominantly originated from the southern and southwestern during this study (Fig. S1a), with an average wind speed of $3.1 \pm 2.5$ m s$^{-1}$. The diurnal variation of local wind direction was controlled by mountain-valley wind, characterized by southern downslope wind in the afternoon with the highest wind speed and a weak northern upslope wind after sunrise until noon (Figs. 2b and S1b). The South Asian summer monsoon advanced into Bay of Bengal on 16 May and covered the entire India on 2 July based on the monsoon report provided by India Meteorological Department (https://imdpune.gov.in/reports.php). The $PM_1$ concentration varied from 0.1 to 12.2 µg m$^{-3}$ with an average value of $1.7 \pm 1.6$ µg m$^{-3}$, which was lower than that at QOMS in the same period in 2016 (4.4 µg m$^{-3}$) (Zhang et al., 2018), and similar to that at Nam Co Station (2.0 µg m$^{-3}$) located in the central TP (Xu et al., 2018). Different from previous study in 2016, which identified several BB emission events with accumulated time more than 10 days (Zhang et al., 2018), we only found a polluted period from 25 April to 1 May with an averaged concentration of 4.6



$\pm$ 2.7 µg m$^{-3}$ influenced by BB emissions (see Sect. 3.2 and 3.5.1) and clean period (1.3 $\pm$ 0.7 µg m$^{-3}$) accounted for nearly 80 % of the entire study (Fig. 2).

Organics was the dominant PM$_1$ species during the study, accounting for 46.2 % on average, followed by sulfate (20.8 %), BC (19.4 %) ammonium (8.5 %), nitrate (4.8 %) and chloride (0.4 %). During the polluted period, the fraction of BC was 24 %, indicating the strong impact of BB emissions. Based on the mass concentrations of inorganic species, the bulk acidity of submicron aerosols was evaluated according to the method in Zhang et al. (2007) and Schueneman et al. (2021), detailed description can be found in Text S2. The PM$_1$ appeared to be generally neutralized in this study, indicated by the scatter plot

between the measured and predicted ammonium in Fig. S2 (Slope = 0.97, R$^2$ = 0.91). This is due to excesses of ammonium in PM$_1$ related to agricultural emissions in South Asia and the underestimate of the predicted ammonium by nitrogen-containing organic compounds from BB emissions (Zhang et al., 2018).



**Figure 2.** Time series of **(a)** temperature and relative humidity (RH), **(b)** wind speed colored by wind direction, **(c)** gaseous SO$_2$ and NH$_3$, **(d)** gaseous NO$_x$ and O$_3$, **(e)** PM$_1$ species, **(f)** mass fractions of PM$_1$ species as well as total PM$_1$ mass concentrations for the entire study. The donut chart shows the average contribution of each species and average PM$_1$ concentration.

### 3.1.2 Diurnal cycles and size distributions of PM$_1$





Figure 3 presents the diurnal variation of PM$_1$ species and meteorological parameters for the entire study. The diurnal cycle

of all species was characterized by the minimum value appearing in the afternoon (~14:00) with the highest air temperature and strongest solar radiation, indicating the dilution effect caused by rapid uplift of atmospheric boundary layer height (ABLH), which could reach 3.5 km above ground level in post-monsoon (Lai et al., 2023). After that, the mountain-valley wind system transformed from northern upslope wind to southern downslope wind. There were two peaks at 8:00 and 18:00, especially for BC, organics and sulfate, consistent with the enhanced southerly wind (Fig. S1b), which could transport

polluted air masses from South Asia. The fraction of PM$_1$ species remained relatively constant during the whole day, as shown in Fig. 3a.

Averaged size distributions of NR-PM$_1$ species for the entire study are shown in Fig. S3. All species peaked at ~640 nm in vacuum aerodynamic diameter ($D_{va}$), indicating the well mixed and aged aerosol particles at QOMS during the sampling period. Moreover, organics exhibited a broad size distribution in smaller size, which might be caused by small particles from

biomass burning (Chen et al., 2022a). As shown in Fig. S3b, organics dominated in small size (< 500 nm, > 60 %), and the contribution decreased with increasing particle size, while inorganic species, especially sulfate, increased accordingly.

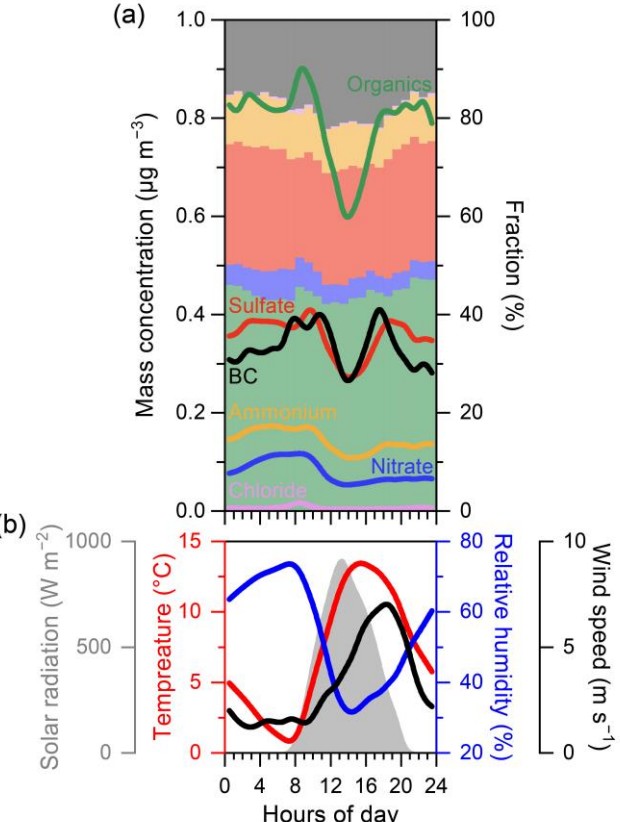

**Figure 3.** Diurnal variations of **(a)** PM$_1$ species and **(b)** meteorological parameters including temperature, relative humidity solar radiation and wind speed.





## 3.2 Source region and transport pathways

To identify the source region of PM$_1$, trajectory cluster analysis and CWT analysis were applied using the HYSPLIT model. The air mass trajectories were classified into three clusters for the entire study period (Fig. 4a), including C1 (from southwest), C2 (from north), C3 (from southeast) contributed to 15.8 %, 10.5 % and 73.7 %, respectively. We further examined the relative contributions of these three clusters during polluted and clean periods (Figs. 4b and c). Distinct difference between the two periods can be observed, i.e., pollution period was dominated by C1 (72.2 %) (Fig. S4), whereas clean period was almost from C3 (87.5 %). Previous studies combining ground-based observation and numerical simulations have shown that QOMS was dominated by southwesterly wind during the pre-monsoon driven by the strong westerly jet aloft and southeasterly wind associated with thermal gradient in the early monsoon before the rainy season (Sun et al., 2017; Sun et al., 2018a). As shown in Fig. S5, the westerlies index was high during the polluted period, and the India-Burma Trough index strengthened in the clean period. In the monsoon transition period, the subtropical jet moves northward causing the weak westerly winds over the Himalayas and then the India-Burma Trough firstly controls further the north, which results in the difference between the two periods in our study. In the polluted period, the westerly drives polluted air mass from northern India and Nepal across the Himalayas. The altitude variations of C1 indeed showed a gradually increase from the ground surface to QOMS (Fig. S6). On the contrary, southeastern Nepal is the main source in the clean period, and the air mass might be even transported from the Bay of Bengal via India-Burma Trough. The high fraction of sulfate in C3 (24 %) indicates the potential marine source (Zhou et al., 2023). The CWT results also suggest different source regions between the two periods and match well with trajectory clusters (Fig. 4d–f). The high potential source area for all species in the entire study is consistent with that in the polluted period, suggesting the important source regions in the northern India and Nepal. In addition, we selected two orbit tracks of CALIPSO satellite passing through QOMS to further investigate the transport of aerosol in the Himalayas (Fig. 1). The vertical distribution inversion of the aerosol extinction coefficient in the polluted period clearly showed aerosols from the foothills climbed over the Himalayas and transported to the interior of the TP. However, the transport process was not captured in the clean period. As shown in Fig. S7, the tropospheric AOD rapidly decreased at ~28 °N in the polluted period and at ~26 °N in the clean period, further supporting the strong aerosol transport from South Asia in the polluted period.





**Figure 4.** The 3-day backward air mass trajectories, trajectory clusters and CWT of PM$_1$ species for the **(a, d)** entire study period, **(b, e)** polluted period and **(c, f)** clean period.

Satellite-observed fire hotspots by the MODIS and AOD from MERRA-2 were used to verify the trajectory analysis above and explain the important contribution from BB emission (Fig. 5). During the polluted period, the trajectories passed through northern India and southwestern Nepal where distributed dense fire hotspots, suggesting serious BB emission. In the clean period, there was basically no trajectory passing through the fire hotspots, even the southwestern Nepal had fewer fire hotspots than those in the polluted period. The distribution of MERRA-2 AOD in the Himalayas also showed latitudinal movement between the two periods (Figs. 5c and d). All AOD contour lines move southward, e.g., contour line with AOD





value of 0.7 (green lines in Figs. 5c and d) moving from central Nepal to southern boundary from the polluted period to clean
210    period, indicating the cleaner atmosphere in source region during the clean period.

**Figure 5.** The distribution of fire hotspots and aerosol optical depth (AOD) around the QOMS as well as 3-day backward air
mass trajectories for the **(a, c)** polluted period and **(b, d)** clean period.

### 3.3 Bulk characteristics of OA

215    The average HRMS and elemental compositions of OA during the study are shown in Fig. 6a. On average, $C_xH_yO_1^+$
dominated the total OA (41 %) followed by $C_xH_yO_2^+$ (28 %), $C_xH_y^+$ (22 %), $H_yO_1^+$ (7 %) and $C_xH_yN_p^+$ (1 %). The two major
oxygenated ion fragments ($C_xH_yO_1^+$ and $C_xH_yO_2^+$) accounted for 69 % of OA at QOMS in this study, which was similar to
those at QOMS in 2016 (66%) (Zhang et al., 2018), Waliguan in the northeastern TP (66%) (Zhang et al., 2019), Mt. Yulong
(68 %) (Zheng et al., 2017), and Nam Co (58%) (Xu et al., 2018), higher than the value at Lhasa as the typical urban city in





the TP (34 %) (Zhao et al., 2022). The organic mass was on average composed of 57 % oxygen, 39 % carbon, and 3 % hydrogen, with an average nominal formula being $C_1H_{1.06}O_{1.1}N_{0.01}$. The average O/C ratio of 1.10 during the entire study was also comparable with those at several sites in the TP (Xu et al., 2024). These similar characteristics of high fraction oxygenated ion fragments and O/C ratio on the TP indicate its overall remoteness and highly oxidized and aged aerosols due to long-range transport. In the polluted period, the O/C ratio was relatively low and H/C ratio was higher (Fig. 6b), suggesting relatively fresh in this period.

Diurnal cycles of O/C and H/C ratios in this study exhibited opposite variations, ranging from 0.88–1.18 and 1.02–1.21, respectively (Fig. 6c). The O/C ratio showed two peaks at ~5:00 and ~19:00, with a pronounced minimum at ~8:00 LT, reflecting alternating influences of aging processes and fresh aerosol transport. Nitrogen oxides ($NO_x$), commonly used as a tracer for anthropogenic and combustion emission (VanderSchelden et al., 2017), peaked at 8:00 and 17:00 (Fig. 6d), and matched with the peak of BC (Fig. 3a) as well as the maximum of southerly wind frequency (Fig. S1b), suggesting the contribution of BB emission from South Asia. Odd-oxygen ($O_x = O_3 + NO_2$), a proxy for photochemical activity (Wang et al., 2017b), and ALWC estimated by the E-AIM model and associated with aqueous-phase SOA production (Sareen et al., 2017), were further used to examine the influence of different chemical processes in diurnal cycles. $O_x$ increased sharply after 8:00, reached its peak around noon, and persisted at elevated levels into the afternoon before rapidly decreasing at night (Fig. 6d). This pattern paralleled the continuous increase of O/C ratio after the impact by BB emission transport in the morning, indicating the importance of photochemical process in OA oxidation at QOMS. The second transport (at 17:00) was not reflected in the element ratios, which is due to the relatively weak intensity and high photochemical conditions. Meanwhile, ALWC reached high levels before dawn, coinciding with the early-morning peak O/C ratio (Fig. 6e). Although ALWC showed the peak at ~8:00, the concurrent influx of BB emissions diluted the aerosol population, leading to a rapid decrease in O/C despite the high aqueous-phase potential.

In addition, we used sulfur oxidation ratio (SOR = $nSO_4^{2-}{}_{(p)}/[nSO_4^{2-}{}_{(p)} + nSO_{2(g)}]$) and nitrogen oxidation ratio (NOR = $nNO_3^{-}{}_{(p)}/[nNO_3^{-}{}_{(p)} + nNO_{2(g)}]$) to evaluate the oxidation degree of their precursors ($SO_2$ and $NO_2$) to secondary inorganic aerosol (SIA) (Fig. 6e), which are closely associated with SOA formation via photochemical or aqueous-phase reactions (Chen et al., 2022b; Hu et al., 2016; Sareen et al., 2017). The diurnal variation of NOR showed two peaks, occurring at 5:00–6:00 and 18:00–19:00, both coinciding with elevated O/C ratios and other oxidation indicators ($O_x$ or ALWC). In contrast, SOR only peaked around 18:00–20:00, indicating that sulfate production was mainly enhanced through photochemical reaction. These results imply that aqueous-phase processes are more effective in promoting nitrogen oxidation, while sulfur oxidation is more strongly influenced by photochemistry at QOMS. Overall, both SOR and NOR prove to be useful oxidation indicators associated with SOA formation, and their roles in different pathways of OA aging will be further elaborated in Sect. 3.5.2.



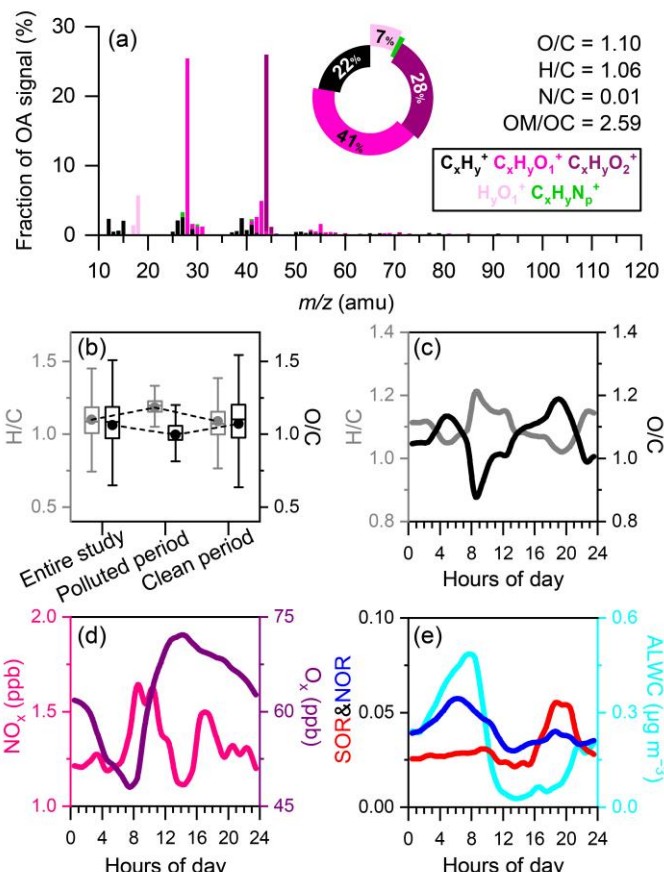

**Figure 6. (a)** The average mass spectrum of organic aerosol, **(b)** the average ratios of H/C and O/C during different periods, as well as diurnal profiles of **(c)** H/C and O/C, **(d)** $NO_x$ and $O_x$, **(e)** SOR, NOR and ALWC.

## 3.4 OA apportionment

255     Source apportionment of OA was performed using PMF analysis on OA HRMS data. Two oxygenated OA (OOA) factors were identified, i.e., a less-oxidized OOA influenced by biomass burning (OOA-BB) and a more-oxidized OOA (MO-OOA). The mass spectra, time series and the corresponding tracer species of the two factors are shown in Fig. 7. The diurnal pattern and fraction of the two factors in different periods are summarized in Fig. 8. The details on each factor are given as follows.





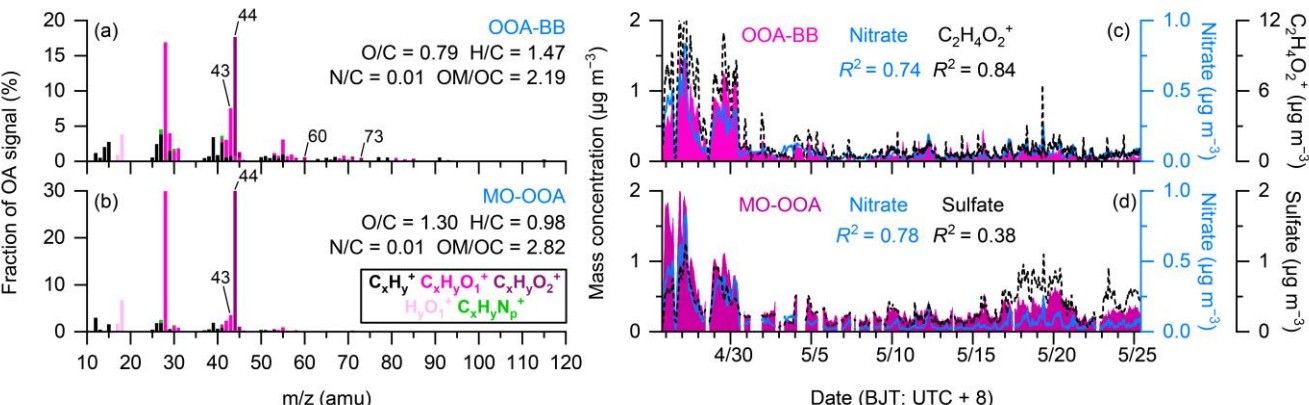

**Figure 7.** High resolution mass spectra (left panel) and time series (right panel) of two organic aerosol factors: **(a, c)** OOA-BB, **(b, d)** MO-OOA. Also shown in the right panel are the corresponding tracer species for comparisons.

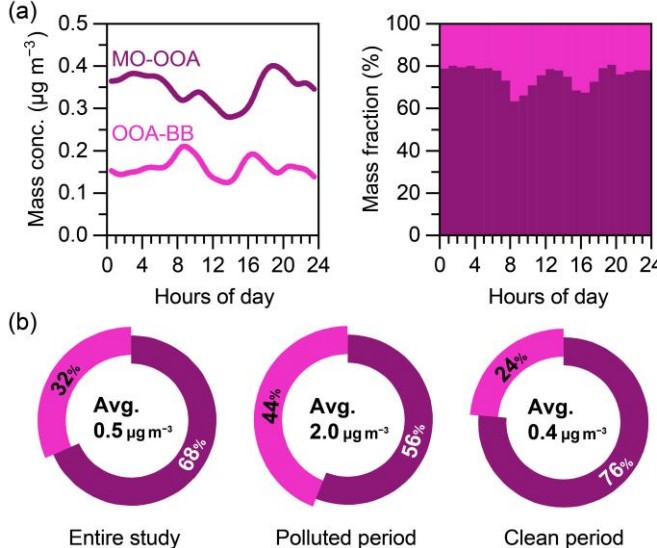

**Figure 8. (a)** Diurnal variations of PMF factor mass concentrations and fraction; **(b)** average fractions of PMF factors in OA during different periods.

### 3.4.1 OOA-BB

The mass spectrum of OOA-BB identified in this study was characterized by a pronounced peak at $m/z$ 44 (dominated by $CO_2^+$) and a relatively higher peak at $m/z$ 43 (mostly $C_2H_3O^+$) compared with MO-OOA (Figs 7a and b). The OOA-BB MS exhibited a high O/C ratio (0.79) and was dominated by $C_xH_yO_1^+$ (41 %), followed by $C_xH_y^+$ (33 %) and $C_xH_yO_2^+$ (20 %) fragments (Fig. S8a). Although its high degree of oxidation, characteristic peaks of $m/z$ 60 and $m/z$ 73 remained evident which are well-known tracers for BB emissions (Alfarra et al., 2007; Cubison et al., 2011). The contribution of the $m/z$ 60





(*f*60) in OOA-BB mass spectra was 0.58 %, exceeding the background threshold of 0.3 % (Cubison et al., 2011), confirming BB origination. The high-resolution mass spectrum of OOA-BB in this study correlated tightly with aged BBOA identified at Waliguan ($R^2$ = 0.97; Fig. S9a) (Zhang et al., 2019), and the less-oxidized OOA (LO-OOA) identified at Nam Co ($R^2$ = 0.88; Fig. S9b) (Xu et al., 2018). The time series of OOA-BB correlated well with $C_2H_4O_2^+$ ($R^2$ = 0.85) and nitrate ($R^2$ = 0.74) (Fig. 7c). These results further highlight its nature of aged BB plumes at QOMS in this period. The average mass concentration of OOA-BB was 0.16 ± 0.26 μg m$^{-3}$ contributing to 29 % of the total OA in the entire study and 41 % in the polluted period. The diurnal profile of OOA-BB showed two peaks, occurring in the morning (8:00–9:00) and afternoon (16:00–17:00), respectively, related to the transport of aged BB plumes aloft, under prevailing southerly winds (Fig. S1b). The afternoon peak also coincided with the photochemistry of the day. In addition, the OOA-BB slightly rose around 4:00–6:00, corresponding to decreased ABLH and nighttime chemistry. Therefore, photochemical reaction and aqueous-phase process could result in the enhancement of OOA-BB during different times (Sect. 3.3).

### 3.4.2 MO-OOA

Another OOA factor, MO-OOA, was characterized by the prominent peak of *m/z* 44 (accounting for 30 % of the total signal) and a lower contribution at *m/z* 43 than OOA-BB. MO-OOA had the highest O/C ratio (1.30) and lowest H/C ratio (0.98) among the identified factors. Its ion composition was dominated by $C_xH_yO_1^+$ (41 %, the same as OOA-BB), but with lower $C_xH_y^+$ (17 %) and higher $C_xH_yO_2^+$ (32 %) contributions (Fig. S8b). The time series of MO-OOA in this study correlated with the SIA species, i.e., nitrate ($R^2$ = 0.78) and sulfate ($R^2$ = 0.38) (Fig. 7d), indicating the aged properties. The MS of MO-OOA in this study resembled those of MO-OOA and low-volatility oxygenated OA (LV-OOA) resolved from previous studies in the TP and other regions (Hu et al., 2016; Sun et al., 2011; Xu et al., 2018; Zhang et al., 2019). MO-OOA had an average concentration of 0.35 ± 0.29 μg m$^{-3}$, accounting for 68 % of the total OA mass. During the clean period, the MO-OOA even increased to 76 % of total OA, 20 % higher than that in the polluted period, indicating the aging and regional background characteristics of the atmospheric aerosol at QOMS during the clean period. The diurnal variation of MO-OOA presented the lowest concentrations around 14:00, similar to OOA-BB, which was driven by the dynamics of the ABLH. MO-OOA enhanced during nighttime and showed a profound peak during afternoon, which corresponded to nighttime chemistry and photochemistry, respectively. The peak of MO-OOA was later than that of OOA-BB, suggesting the evolution from OOA-BB to MO-OOA.

### 3.5 OA evolution

### 3.5.1 Ageing process of OA

To investigate the ageing process of OA and potential BB emission during the different periods, we employed triangle plots of *f*44 vs. *f*43 and *f*44 vs. *f*60 of bulk OA (Cubison et al., 2011; Ng et al., 2010), as well as Van Krevelen diagram of elemental ratios (Heald et al., 2010) (Fig. 9). The dataset during the polluted period and clean period in *f*44 vs. *f*60 space are





totally different. In the polluted period, a considerable number of data points have high $f60$ value ($> 0.3\%$) associated with high $f44$ values ($> 0.2$), indicating the aerosols are aged BBOA. In addition, BC concentrations decreased with increasing $f44$ and decreasing $f60$ (Fig. 9a), further proving the importance of BB emission in OA evolution at QOMS. The data points

305   during the clean period are almost in the non-BB influence region with low $f60$ ($< 0.3\%$) (Fig. 9b), and there is no significant variation trend of BC concentrations in the $f44$ vs. $f60$ space.

In the $f44$ vs. $f43$ space, the data points during two periods show similar oxidation pathways, moving towards the top-left direction (higher $f44$ and lower $f43$) (Fig.9c and d). The OOA-BB shows an obvious reduction with increasing $f44$ values, indicating the oxidation process from relatively fresh OA (OOA-BB) to aged OA (MO-OOA) in this study. The slopes of

310   linear fitting of the data points in different period in the Van Krevelen diagram were close in the polluted period ($-0.60$) and clean period ($-0.64$) (Fig. 9e), suggesting the overall oxidation pathways with the addition of carboxylic acid with fragmentation (Heald et al., 2010; Ng et al., 2011). The slope in this study was similar to those in the TP such as Waliguan ($-0.64$) (Zhang et al., 2019) and Nam Co ($-0.76$) (Xu et al., 2018), as well as the data set of eight field measurements in the TP ($-0.66$) (Xu et al., 2024).

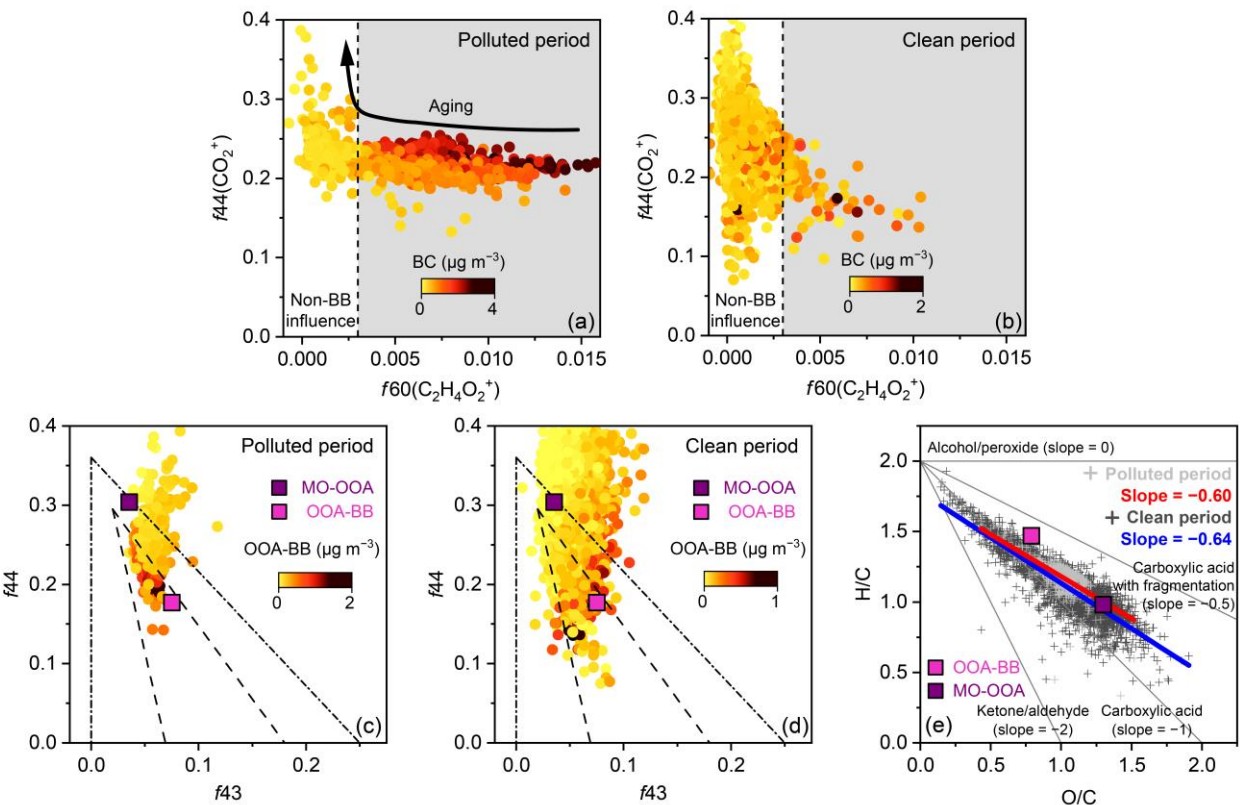

315

**Figure 9.** $f44$ versus $f60$ and $f44$ versus $f43$ during **(a, c)** polluted period and **(b, d)** clean period, as well as **(e)** Van Krevelen (VK) diagram of OA. The dashed lines and dashed-dotted lines in c–d refer to triangular regions that encompass ambient OOA (Ng et al., 2010) and laboratory measurements OOA (Lambe et al., 2011), respectively.




### 3.5.2 Formation mechanism of OOA during polluted period

To explore the potential formation mechanism of OA during long-range transport, the polluted period in this study was dedicated in detail coinciding the formation of SIA and other indexes. OOA-BB, MO-OOA, and ΣOOA (i.e., OOA-BB + MO-OOA) production all showed positive relationships with $O_x$, SOR, ALWC, and NOR (Table S1). Meanwhile, $\Delta OOA/\Delta O_x$ (linear regression slopes of OOA PMF factors versus $O_x$) and $\Delta OOA/\Delta SOR$ had enhancements in the daytime (Fig. 10a), highlighting the dominant role of photochemical reaction in OOA production. In contrast, $\Delta OOA/\Delta ALWC$ and $\Delta OOA/\Delta NOR$ increased in the nighttime, further suggesting aqueous-phase processes played more important roles under higher relative humidity and nighttime. Previous studies also found that SOR and NOR were consistent with OA aging processes (Chen et al., 2022b; Zhang et al., 2021), but the specific processes coupled with them are not necessarily identical across regions, which could be associated with the different atmospheric conditions and SOA precursors.

As shown in Fig. 10b–d, aqueous-phase process was an important contributor to the OOA production, based on the fact that concentrations of OOA factors had tightly positive correlation with ALWC and increase with NOR. In addition, $NH_3$ also showed a similar trend with OOA concentrations. Lv et al. (2022) reported that $NH_3$ can significantly elevate ALWC and pH through the formation of $NH_4NO_3$ and neutralization with organic acids, resulting in SOA production by ALW formation and WSOC partitioning from gas-phase to aerosol-phase, suggesting a promoting effect of $NH_3$ on OOA production at QOMS. OOA production was also controlled by photochemical reaction at QOMS (Fig 10e–g). We found strong correlation between ΣOOA and $O_x$ with a linear regression slope of 0.04 µg m$^{-3}$ ppb$^{-1}$, which was similar to the ratio at Houston (0.03 µg m$^{-3}$ ppb$^{-1}$) (Wood et al., 2010), lower than those at Tokyo (0.19 µg m$^{-3}$ ppb$^{-1}$) (Morino et al., 2014), Paris (0.14 µg m$^{-3}$ ppb$^{-1}$) (Zhang et al., 2015a) and Beijing (0.12–1.2 µg m$^{-3}$ ppb$^{-1}$) (Hu et al., 2016), and located within the range of 0.02–0.17 µg m$^{-3}$ ppb$^{-1}$ in New York City (Rogers et al., 2025). Although the $\Delta OOA/\Delta O_x$ slope at QOMS was almost less than those in urban areas strongly influenced by anthropogenic emissions, the value cannot be ignored in this relatively pristine area with low aerosol mass loading, indicating the photochemical pollution remains important for air quality here. The SOR was also consistent with the variation trend of OOA factors. Therefore, aqueous-phase process and photochemical reaction together dominate OOA formation at QOMS, and elevate OOA concentrations at different times of the day respectively.





**Figure 10. (a)** Linear regression slopes of ΔOOA/ΔO$_x$, ΔOOA/ΔSOR, ΔOOA/ΔALWC, ΔOOA/ΔNOR for OOA factors grouped by daytime (solar radiation > 0) and nighttime (solar radiation = 0). **(b–d)** OOA factors plotted against ALWC colored by NH$_3$, and sized by NOR. **(e–g)** OOA factors plotted against O$_x$ colored by SOR. "ns" represents the slope is not significantly non-zero at $p = 0.05$.

### 3.5.3 Diurnal formation processes of MO-OOA based on a box model



To further capture the diurnal formation processes of MO-OOA, a box model was utilized in this study to elucidate the details in this process (Chen et al., 2021), which includes the direct emission, chemical reaction, depositional loss, horizontal advection, and vertical transport (the primitive equation Eq. S2 can be found in Text S3). After considering the following conditions, (1) MO-OOA has no direct emission sources as an aged SOA factor, (2) the effect of horizontal advection is negligible due to multiple diurnal cycles, and (3) the aging of OOA-BB is considered as a source of MO-OOA and the further oxidation of MO-OOA is a sink, the formation rate of MO-OOA is determined by the following equation:

$$\frac{dc_{\text{MO-OOA}}}{dt} = \left(k_1 c_{\text{OOA-BB}} - k_2 c_{\text{MO-OOA}}\right) c_{\text{OH}} - \frac{v_{d_M} c_{\text{MO-OOA}}}{H(t)} + \frac{1}{H(t)} \frac{dH}{dt}\bigg|_{dH/dt>0} \left(c^{a}_{\text{MO-OOA}} - c_{\text{MO-OOA}}\right) \tag{3}$$

The reaction rates $k_1$ and $k_2$ were set to the same as those of Chen et al. (2021), which used $1 \times 10^{-12}$ and $1 \times 10^{-13}$ cm³ molecule⁻¹ s⁻¹ for LO-OOA and MO-OOA. The $v_{d_M}$ is maximum deposition velocity, set to be 13.2 mm s⁻¹ according to the maximum aerosol deposition velocity (0.76 μm was chosen to align with the peak of ~0.64 μm in this study) in a valley of the Italian Alps in summer (Urgnani et al., 2022). The $H(t)$ is the ABLH, derived from ERA5 datasets over the same period. The OH concentrations measured at Nam Co were used to estimate the OH concentration at QOMS (Wang et al., 2023). Due to the impact of South Asia, a factor of 10 was used at QOMS, where the OH concentration is higher in urban and industrial areas (Li et al., 2012). The MO-OOA aloft concentration ($c^{a}_{\text{MO-OOA}}$) was considered altitude-dependent and calculated via correlating the observed MO-OOA concentration with PBLH (Fig. S10).

As shown in Fig.11, this box model well captured the peak of MO-OOA at 18:00–19:00 associated with photochemical reaction, and the minimum at 8:00 resulting from high deposition loss. However, the enhancement by aqueous-phase process at midnight and the dilution effect of ABLH at noon are ignored, due to the parameter limitation of this box model. The high contribution of mixing from aloft to formation rate is a misinterpretation caused by transport (Fig. 11a). The model is able to describe the general diurnal trend of MO-OOA, but more parameters are needed to improve the details in atmospheric physical and chemical processes.

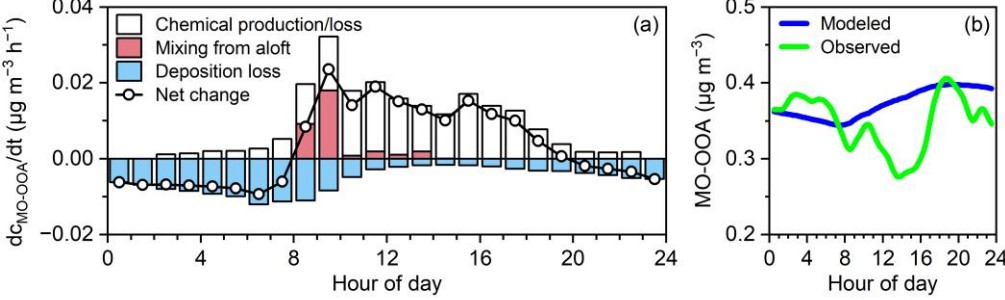

**Figure 11.** Diurnal profiles of **(a)** simulated contributions from different processes and the net change rate of MO-OOA as well as **(b)** observed and modeled MO-OOA concentration.

## 4 Conclusions



Intensive campaigns at QOMS in 2022 provided new insight into the physicochemical properties and aging process of
submicron aerosols in the Himalayas. The average mass concentration of $PM_1$ was low ($1.7 \pm 1.6$ μg m$^{-3}$), with OA as the
dominant component (46.2 %). Source analyses indicated that polluted periods were strongly influenced by BB plumes
transported from South Asia, while clean periods reflected more regional background conditions. The diurnal pattern of OA
elemental compositions and oxidation indicators indicated the potential contributions of photochemical reaction and
aqueous-phase process. PMF analysis performed on the OA HRMS resolved two OA factors, including OOA-BB and MO-
OOA. The OOA formation at QOMS was dominated via aqueous-phase process and photochemical reaction in nighttime
and daytime respectively, consistent with the SOR and NOR as the different oxidation processes tracers. The box model also
suggested that the photochemical reaction played an important role in MO-OOA formation. Overall, this study highlights the
critical role of long-range transport and multiphase oxidation in shaping OA composition in the Himalayas. These findings
advance our understanding of aerosol aging processes in remote high-altitude regions and provide a basis for improving the
representation of OA sources and transformations in regional and global models.

## Data availability

The data in this study are available upon request from the corresponding author.

## Author contributions

JX designed the study. YW analyzed the data and wrote the manuscript. YA, JX, and SH organized and supervised the field
measurement campaigns. YT and KL conducted the field measurements. All authors reviewed and commented on the final
form of the manuscript.

## Competing interests

The authors declare that they have no conflict of interest.

## Financial support

This study was supported by the National Natural Science Foundation of China (42476249 and 42021001), the National Key
Research and Development Program of China (2023YFF0805300), and the Fundamental Research Funds for the Central
Universities.

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
