# Peer review of "Highly time-resolved chemical characteristics and aging process of submicron aerosols over the central Himalayas"

_EGUsphere, 2025_

## Author Comment (AC1)

We appreciate the reviewers for their valuable comments and suggestions for our manuscript. We have revised our manuscript accordingly and our response point-by-point are presented below. Note that the comments are in **black**, and our responses are in **blue**, while the corresponding revise in the manuscript are in **red**.

**Response to Reviewer #1**

The manuscript provides valuable insights into the chemical composition and aging processes of submicron aerosols in the Himalayas, particularly focusing on the impact of biomass burning (BB) aerosols during the pre-monsoon period. The use of high-resolution mass spectrometry along with real-time gas analyzers offers a comprehensive view of aerosol characteristics and their formation mechanisms. Additionally, the combination of positive matrix factorization and air mass trajectory analyses provides a clear understanding of the sources and transport pathways of these aerosols. The manuscript is generally well written, and I recommend it for publication after addressing the following comments:

Thank you sincerely for your constructive and positive comments. We have carefully revised our manuscript point-by-point.

1. The authors used default RIEs for mass quantification. Did the authors perform any calibration of the AMS? It should be possible to obtain the actual RIE for NH4 from the calibration data.

The field campaign, especially using real-time instruments, in the Himalayas is significantly constrained by the harsh field conditions. Based on our previous field studies using the same AMS in the TP, the RIE for ammonium is generally stable and close to the default value (< 5 %) (Xu et al., 2018; Xu et al., 2024; Zhang et al., 2018). Although the calibration of the AMS was not performed in this study due to the harsh field conditions, it is reasonable to use the default value of ammonium RIE based on our previous studies in the TP.

Xu, J., Zhang, Q., Shi, J., Ge, X., Xie, C., Wang, J., Kang, S., Zhang, R., and Wang, Y.: Chemical characteristics of submicron particles at the central Tibetan Plateau: insights from aerosol mass spectrometry, Atmos. Chem. Phys., 18, 427–443, https://doi.org/10.5194/acp-18-427-2018, 2018.
Xu, J., Zhang, X., Zhao, W., Zhai, L., Zhong, M., Shi, J., Sun, J., Liu, Y., Xie, C., Tan, Y., Li, K., Ge, X., Zhang, Q., and Kang, S.: High-resolution physicochemical dataset of atmospheric aerosols over the Tibetan Plateau and its surroundings, Earth Syst. Sci. Data, 16, 1875–1900, https://doi.org/10.5194/essd-16-1875-2024, 2024.
Zhang, X., Xu, J., Kang, S., Liu, Y., and Zhang, Q.: Chemical characterization of long-range transport biomass burning emissions to the Himalayas: insights from high-resolution aerosol mass spectrometry, Atmos. Chem. Phys., 18, 4617–4638, https://doi.org/10.5194/acp-18-4617-2018, 2018.

2. Regarding the diurnal cycles, could the authors check those during the clean periods? The reason for this is that the sources and processes in the clean and polluted periods are different. The diurnal cycle for the entire study could be primarily influenced by the first polluted period with high mass loadings. Similarly, I suggest the authors examine the contributions of different processes in the two distinct periods (Figure 11).

We separately analyzed the diurnal cycles of PM$_1$ species, elemental ratios, and related indicators during the clean period and polluted period. The results show that the diurnal patterns are largely similar between the two periods, with only minor differences. The overall diurnal cycle is primarily influenced by the clean period, which constitutes 80 % of the entire study. Firstly, diurnal cycles of all PM$_1$ species were characterized by the minimum value appearing in the afternoon and two peaks at 8:00 and 18:00. The two peaks match well with NO$_x$ (grey shadows), indicating the transport. Secondly, the variations of elemental ratios are consistent with oxidation indicators (yellow shadows). The high ALWC and NOR at 5:00–6:00 indicates aqueous-phase reactions and the SOR and O$_x$ peak at 18:00–19:00 suggesting photochemical processes. The difference is the second transport (at 17:00) was not reflected in the element ratios in the entire period and clean period, while element ratios (high H/C and low O/C) matched with second transport during the polluted period, which is due to the relatively weak intensity and high photochemical conditions in the clean period. Accordingly, we have added Figure S3 in the Supplement to provide a detailed comparison of the diurnal cycles in the two periods.

"There were two peaks at 8:00 and 18:00, especially for BC, organics and sulfate, consistent with the enhanced southerly wind (Fig. S1b), which could transport polluted air masses from South Asia. The two peaks were also identified separately in the clean and polluted periods with the similar diurnal pattern (Fig. S3)." (Lines 163–166 in the main text)

"The second transport (at 17:00) was not reflected in the element ratios, which is due to the relatively weak intensity and high photochemical conditions in the entire period. The element ratios matched with second transport during the polluted period with strong intensity (Fig. S3)." (Lines 237–239 in the main text)

"The diurnal variation of NOR showed two peaks, occurring at 5:00–6:00 and 18:00–19:00, both coinciding with elevated O/C ratios and other oxidation indicators (O$_x$ or ALWC). In contrast, SOR only peaked around 18:00–20:00, indicating that sulfate production was mainly enhanced through photochemical reaction. These results imply that aqueous-phase processes are more effective in promoting nitrogen oxidation, while sulfur oxidation is more strongly influenced by photochemistry at QOMS. The SOR and NOR had more obvious peaks in polluted period (Fig. S3), suggesting the stronger oxidation processes. Overall, both SOR and NOR prove to be useful oxidation indicators associated with SOA formation, and their roles in different pathways of OA aging during polluted period will be further elaborated in Sect. 3.5.2." (Lines 246–253 in the main text)

[Figure]

**Figure S3.** Diurnal variations of PM₁ species, elemental ratios, gas indicators, SOR, NOR and ALWC during the entire study, polluted period and clean period. The grey shadows mark the potential transport periods with peaks of PM₁ species and NOₓ. The yellow shadows mark the oxidation processes.

We also agree with the reviewer that the box model should be performed in the two distinct periods. In this box-model, the contribution of mixing from aloft is based on the linear fitting between PBLH and MO-OOA (Chen et al., 2021). However, we found there is no correlation between the two parameters in the clean period when we tried to apply the model in the two distinct periods, which

means the box-model fails to meet the application requirements in the clean period and entire period (including clean period). Therefore, we only used the box model in the polluted period, which is seriously affected by biomass burning emission. The results show that box model successfully captured the increase of MO-OOA associated with photochemical reaction, and the minimum at 8:00 resulting from high deposition loss. The detailed corresponding revise in the manuscript can be found below.

Chen, Y., Guo, H., Nah, T., Tanner, D.J., Sullivan, A.P., Takeuchi, M., Gao, Z., Vasilakos, P., Russell, A.G., Baumann, K., Huey, L.G., Weber, R.J., and Ng, N.L.: Low-molecular-weight carboxylic acids in the southeastern US: formation, partitioning, and implications for organic aerosol aging, Environ. Sci. Technol., 55, 6688–6699, https://doi.org/10.1021/acs.est.1c01413, 2021.

**"3.5.3 Diurnal formation processes of MO-OOA based on a box model**

To further capture the diurnal formation processes of MO-OOA, a box model was utilized in the polluted period to elucidate the details in these processes (Chen et al., 2021), which includes the direct emission, chemical reaction, depositional loss, horizontal advection, and vertical transport (the primitive equation Eq. S2 can be found in Text S3). After considering the following conditions, (1) MO-OOA has no direct emission sources as an aged SOA factor, (2) the effect of horizontal advection is negligible due to multiple diurnal cycles, and (3) the aging of OOA-BB is considered as a source of MO-OOA and the further oxidation of MO-OOA is a sink, the formation rate of MO-OOA is determined by the following equation:

$$\frac{dc_{\text{MO-OOA}}}{dt} = \left(k_1 c_{\text{OOA-BB}} - k_2 c_{\text{MO-OOA}}\right) c_{\text{OH}} - \frac{v_{d_M} c_{\text{MO-OOA}}}{H(t)} + \frac{1}{H(t)} \frac{dH}{dt}\bigg|_{dH/dt>0} \left(c_{\text{MO-OOA}}^a - c_{\text{MO-OOA}}\right) \quad (1)$$

The reaction rates $k_1$ and $k_2$ were set to the same as those of Chen et al. (2021), which used $1 \times 10^{-12}$ and $1 \times 10^{-13}$ cm$^3$ molecule$^{-1}$ s$^{-1}$ for LO-OOA and MO-OOA. The $v_{d_M}$ is maximum deposition velocity, set to be 13.2 mm s$^{-1}$ according to the maximum aerosol deposition velocity (0.76 μm was chosen to align with the peak of ~0.64 μm in this study) in a valley of the Italian Alps in summer (Urgnani et al., 2022). The $H(t)$ is the ABLH, derived from ERA5 datasets over the same period. The OH concentrations measured at Nam Co were used to estimate the OH concentration at QOMS (Wang et al., 2023). Due to the impact of South Asia, a factor of 3 was used at QOMS, where the OH concentration is higher in urban and industrial areas (Li et al., 2012). The MO-OOA aloft concentration ($c_{\text{MO-OOA}}^a$) was considered altitude-dependent and calculated via correlating the observed MO-OOA concentration with PBLH (Fig. S10).

As shown in Fig.11, this box model well captured the increase of MO-OOA associated with photochemical reaction, and the minimum at 8:00 resulting from high deposition loss. However, the enhancement by aqueous-phase process at midnight and the dilution effect of ABLH at noon are ignored, due to the parameter limitation of this box model. The model is able to describe the general diurnal trend of MO-OOA, but more parameters are needed to improve the details in atmospheric physical and chemical processes." (Lines 352–372 in the main text)

[Figure]

**Figure 11.** Diurnal profiles of (a) simulated contributions from different processes and the net change rate of MO-OOA as well as (b) observed and modeled MO-OOA concentration in the polluted period.

[Figure]

**Figure S10.** MO-OOA concentration as a function of ABLH in the polluted period.

3. The SOR is notably low compared to previous studies. It is somewhat surprising that the SO2 mixing ratio is relatively high (~3 ppb on average) in this study. Could this be a real measurement, or might it be influenced by instrument uncertainties? It's like a baseline there?

Upon careful re-examination of the raw $SO_2$ data, we discovered that a calibration factor has been omitted, leading to an overestimation. Now we display the calibrated $SO_2$ data in Fig. 2c, and the timeseries shows the obvious diurnal variation. Due to the sparse observations over southern Tibetan plateau, there is few field works reported $SO_2$ data. The averaged value (0.86 ppb) in this study is consistent with that from previous study in the northeastern TP (Waliguan, <1 ppb) (Ma et al., 2020). We also compared our data with "ChinaHighSO$_2$" dataset, a 1-km resolution dataset of $SO_2$ for China generated from big data sources using artificial intelligence (Wei et al., 2023). The monthly $SO_2$ concentration at QOMS in May 2022 is 9.8 μg m$^{-3}$ (~3.4 ppb). This is obviously higher than the result of our field measurement, which could be due to high uncertainty of the dataset in remote areas.

The corresponding SOR value is recalculated based on the calibrated $SO_2$ data and ranges from 0.005 to 0.9 with an average of 0.1. Cheng et al. (2016) found the concentration ratios of sulfate to sulfur dioxide increase as $PM_{2.5}$ levels increase, and concentration ratios are ~0.1 during clean to moderately polluted conditions, which is consistent with our results.

Although the $SO_2$ and SOR values are recalculated, their variation trends during this study have not changed, which the conclusions are not affected.

Cheng, Y., Zheng, G., Wei, C., Mu, Q., Zheng, B., Wang, Z., Gao, M., Zhang, Q., He, K., Carmichael, G., Poschl, U., and Su, H.: Reactive nitrogen chemistry in aerosol water as a source of sulfate during haze events in China, Science Advances, 2, e1601530, https://doi.org/10.1126/sciadv.1601530, 2016.

Ma, J., Doerner, S., Donner, S., Jin, J., Cheng, S., Guo, J., Zhang, Z., Wang, J., Liu, P., Zhang, G., Pukite, J., Lampe, J., and Wagner, T.: MAX-DOAS measurements of $NO_2$, $SO_2$, HCHO, and BrO at the Mt. Waliguan WMO GAW global baseline station in the Tibetan Plateau, Atmos. Chem. Phys., 20, 6973–6990, https://doi.org/10.5194/acp-20-6973-2020, 2020.

Wei, J., Li, Z., Wang, J., Li, C., Gupta, P., and Cribb, M.: Ground-level gaseous pollutants ($NO_2$, $SO_2$, and CO) in China: daily seamless mapping and spatiotemporal variations, Atmos. Chem. Phys., 23, 1511–1532, https://doi.org/10.5194/acp-23-1511-2023, 2023.

[Figure]

**Figure 2.** Time series of (a) temperature and relative humidity (RH), (b) wind speed colored by wind direction, (c) gaseous $SO_2$ and $NH_3$, (d) gaseous $NO_x$ and $O_3$, (e) $PM_1$ species, (f) mass fractions of $PM_1$ species as well as total $PM_1$ mass concentrations for the entire study. The donut chart shows the average contribution of each species and average $PM_1$ concentration.

4. For Figure 9, could the authors clarify the labels in the first two plots? Specifically, is it f44 or f(CO2+), and f60 or f(C2H4O2+)? These two have some differences.

We update Figure 9, and the labels in the first two plots are *f*44 and *f*60.

[Figure]

**Figure 9.** *f*44 versus *f*60 and *f*44 versus *f*43 during (a, c) polluted period and (b, d) clean period, as well as (e) Van Krevelen (VK) diagram of OA. The dashed lines and dashed-dotted lines in c–d refer to triangular regions that encompass ambient OOA (Ng et al., 2010) and laboratory measurements OOA (Lambe et al., 2011), respectively.

**Response to Reviewer #2**

In this study, an intensive field campaign was conducted at the Qomolangma Station for Atmospheric and Environmental Observation and Research (QOMS) on the northern slope of the Himalayas (4276 m a.s.l.), during the pre-monsoon period, using HR-ToF-AMS and gas analyzers to study the organic aerosol (OA) oxidation pathways and secondary OA (SOA) formation mechanism under the influence of biomass burning emission. The authors found two periods with different source regions based on HYSPLIT back trajectories and concentration-weighted trajectory (CWT) analysis. Using MODIS fire hotspots and aerosol vertical distributions from CALIPSO, the atmosphere at QOMS was influenced by biomass burning plumes from South Asia in the polluted period. Source apportionment of OA identified two relatively oxidized OAs, a less-oxidized OOA influenced by biomass burning (OOA-BB) and a more-oxidized OOA (MO-OOA). Aqueous-phase process and photochemical reaction together elevated OOA concentrations and ageing processing, consistent with secondary inorganic aerosol production indicators (SOR and NOR). The photochemical reaction was confirmed to be an important contributor to MO-OOA formation via a box model. Overall, the dataset including synchronous AMS and trace gas results provided by this work is valuable. The manuscript is well written and the topic fits well in the scope of ACP. I recommend this manuscript can be published after some minor points.

Thank you sincerely for your constructive and positive comments. We have carefully revised our manuscript and addressed all the comments point-by-point.

1. Section 3.2: Were the trajectories calculated and clustered during the entire study, or independently in the clean period and polluted period as shown in Fig. 4?

We calculated and clustered the HYSPLIT trajectories during the entire study. In Fig. 4a–c, we show the same three clusters identified during the entire study, and cluster contributions in different periods are based on the clusters identified during the entire study. We have clarified in the text in Lines 178–179, "The air mass trajectories were classified into three clusters for the entire study period (Fig. 4a), including C1 (from southwest), C2 (from north) and C3 (from southeast) contributed to 15.8 %, 10.5 % and 73.7 %, respectively. We further examined the relative contributions of these three clusters during polluted and clean periods (Figs. 4b and c)."

2. The average OA concentration after PMF analysis in Figure 8 is 0.5 μg m$^{-3}$, which is obviously lower than the value in the abstract "46.2% of 1.7 ± 1.6 μg m$^{-3}$". Did the authors apply RIE and CE after PMF analysis? Or the contribution of residual factor in PMF result is very high?

After we carefully examined the PMF processes, the RIE and CE were not applied to the OA factors. Now we correct the results in Fig. 8, Fig. 10, Fig. 11, Fig. S10 and Table S1 which include the OA factor information, as well as the OOA-BB and MO-OOA average concentrations in sections 3.4.1 and 3.4.2. The slope of OOA and O$_x$ in section 3.5 is also corrected.

"The average mass concentration of OOA-BB was 0.23 ± 0.36 μg m$^{-3}$ contributing to 32 % of the total OA in the entire study and 44 % in the polluted period." (Lines 278–280 in the main text)

"MO-OOA had an average concentration of 0.50 ± 0.42 μg m$^{-3}$, accounting for 68 % of the total OA

mass." (Lines 292–293 in the main text)

"We found strong correlation between ΣOOA and Ox with a linear regression slope of 0.06 µg m$^{-3}$ ppb$^{-1}$, which was similar to the ratio at Houston (0.03 µg m$^{-3}$ ppb$^{-1}$) (Wood et al., 2010), lower than those at Tokyo (0.19 µg m$^{-3}$ ppb$^{-1}$) (Morino et al., 2014), Paris (0.14 µg m$^{-3}$ ppb$^{-1}$) (Zhang et al., 2015a) and Beijing (0.12–1.2 µg m$^{-3}$ ppb$^{-1}$) (Hu et al., 2016), and located within the range of 0.02–0.17 µg m$^{-3}$ ppb$^{-1}$ in New York City (Rogers et al., 2025)." (Lines 337–341 in the main text)

3. Line 10: replace "during 25 April 2022 to…" with "from 25 April 2022 to…".

Revised as the reviewer suggested.

"An Aerodyne high-resolution time-of-flight aerosol mass spectrometer co-located with gas analyzers was deployed from 25 April 2022 to 25 May 2022 to study the highly time-resolved chemical characteristics and aging process of submicron aerosols (PM$_1$) on the northern slope of the Himalayas." (Lines 9–12 in the main text)

4. Line 146: replace "BC (19.4 %) ammonium (8.5 %)" with "BC (19.4 %), ammonium (8.5 %)".

Revised as the reviewer suggested.

"Organics was the dominant PM$_1$ species during the study, accounting for 46.2 % on average, followed by sulfate (20.8 %), BC (19.4 %), ammonium (8.5 %), nitrate (4.8 %) and chloride (0.4 %)." (Lines 145–146 in the main text)

5. Line 179: replace "C2 (from north), C3 (from southeast)" with "C2 (from north) and C3 (from southeast)".

Revised as the reviewer suggested.

"The air mass trajectories were classified into three clusters for the entire study period (Fig. 4a), including C1 (from southwest), C2 (from north) and C3 (from southeast) contributed to 15.8 %, 10.5 % and 73.7 %, respectively." (Lines 178–179 in the main text)

6. Line 188: replace "gradually" with "gradual".

Revised as the reviewer suggested.

"The altitude variations of C1 indeed showed a gradual increase from the ground surface to QOMS (Fig. S6)." (Lines 189–190 in the main text)

7. Line 269: replace "Although its high degree of oxidation" with "Although its oxidation degree was high".

Revised as the reviewer suggested.

"Although its oxidation degree was high, characteristic peaks of *m/z* 60 and *m/z* 73 remained evident which are well-known tracers for BB emissions (Alfarra et al., 2007; Cubison et al., 2011)." (Lines 272–273 in the main text)

8. Line 310: replace "in different period" with "in different periods".

Revised as the reviewer suggested.

"The slopes of linear fitting of the data points in different periods in the Van Krevelen diagram were close in the polluted period (−0.60) and clean period (−0.64) (Fig. 9e), suggesting the overall oxidation pathways with the addition of carboxylic acid with fragmentation (Heald et al., 2010; Ng et al., 2011)." (Lines 312–315 in the main text)

9. Line 330: replace "increase with NOR" with "increased with NOR".

Revised as the reviewer suggested.

"As shown in Fig. 10b–d, aqueous-phase process was an important contributor to the OOA production, based on the fact that concentrations of OOA factors had tightly positive correlation with ALWC and increased with NOR." (Lines 332–333 in the main text)

10. Line 334: replace "Fig 10e–g" with "Fig. 10e–g".

Revised as the reviewer suggested.

"OOA production was also controlled by photochemical reaction at QOMS (Fig. 10e–g)." (Line 337 in the main text)

11. Line 350: replace "in this process" with "in these processes".

Revised as the reviewer suggested.

"To further capture the diurnal formation processes of MO-OOA, a box model was utilized in the polluted period to elucidate the details in these processes (Chen et al., 2021), which includes the direct emission, chemical reaction, depositional loss, horizontal advection, and vertical transport (the primitive equation Eq. S2 can be found in Text S3)." (Lines 353–355 in the main text)

12. Lines 374-385: polluted period and clean period? This study only identified one polluted period and one clean period.

Revised as the reviewer suggested.

"Source analyses indicated that the polluted period was strongly influenced by BB plumes transported from South Asia, while the clean period reflected more regional background conditions." (Lines 379–380 in the main text)